# The Impact of Time between Booster Doses on Humoral Immune Response in Solid Organ Transplant Recipients Vaccinated with BNT162b2 Vaccines

**DOI:** 10.3390/v16060860

**Published:** 2024-05-28

**Authors:** Sebastian Rask Hamm, Josefine Amalie Loft, Laura Pérez-Alós, Line Dam Heftdal, Cecilie Bo Hansen, Dina Leth Møller, Mia Marie Pries-Heje, Rasmus Bo Hasselbalch, Kamille Fogh, Annemette Hald, Sisse Rye Ostrowski, Ruth Frikke-Schmidt, Erik Sørensen, Linda Hilsted, Henning Bundgaard, Peter Garred, Kasper Iversen, Michael Perch, Søren Schwartz Sørensen, Allan Rasmussen, Caroline A. Sabin, Susanne Dam Nielsen

**Affiliations:** 1Viro-Immunology Research Unit, Department of Infectious Diseases, Rigshospitalet, Copenhagen University Hospital, 2100 Copenhagen, Denmark; 2Laboratory of Molecular Medicine, Department of Clinical Immunology, Section 7631, Rigshospitalet, Copenhagen University Hospital, 2200 Copenhagen, Denmark; 3Department of Cardiology, Rigshospitalet, Copenhagen University Hospital, 2100 Copenhagen, Denmark; 4Department of Emergency Medicine, Herlev and Gentofte Hospital, Copenhagen University Hospital, 2730 Herlev, Denmark; 5Department of Cardiology, Herlev and Gentofte Hospital, Copenhagen University Hospital, 2730 Herlev, Denmark; 6Department of Clinical Immunology, Section 2034, Rigshospitalet, Copenhagen University Hospital, 2100 Copenhagen, Denmark; 7Department of Clinical Medicine, Faculty of Health and Medical Sciences, University of Copenhagen, 2200 Copenhagen, Denmark; 8Department of Clinical Biochemistry, Rigshospitalet, Copenhagen University Hospital, 2100 Copenhagen, Denmark; 9Department of Nephrology, Rigshospitalet, Copenhagen University Hospital, 2100 Copenhagen, Denmark; 10Department of Surgical Gastroenterology, Rigshospitalet, Copenhagen University Hospital, 2100 Copenhagen, Denmark; 11Centre for Clinical Research, Epidemiology, Modelling and Evaluation, Institute for Global Health, UCL, Royal Free Campus, Rowland Hill St, London NW3 2PF, UK

**Keywords:** solid organ transplant recipient, COVID-19, vaccine, booster, SARS-CoV-2, BNT162b2, immunogenicity

## Abstract

As solid organ transplant (SOT) recipients remain at risk of severe outcomes after SARS-CoV-2 infections, vaccination continues to be an important preventive measure. In SOT recipients previously vaccinated with at least three doses of BNT162b2, we investigated humoral responses to BNT162b2 booster doses. Anti-SARS-CoV-2 receptor binding domain (RBD) immunoglobulin G (IgG) was measured using an in-house ELISA. Linear mixed models were fitted to investigate the change in the geometric mean concentration (GMC) of anti-SARS-CoV-2 RBD IgG after vaccination in participants with intervals of more or less than six months between the last two doses of vaccine. We included 107 SOT recipients vaccinated with a BNT162b2 vaccine. In participants with an interval of more than six months between the last two vaccine doses, we found a 1.34-fold change in GMC per month (95% CI 1.25–1.44), while we found a 1.09-fold change in GMC per month (95% CI 0.89–1.34) in participants with an interval of less than six months between the last two vaccine doses, resulting in a rate ratio of 0.82 (95% CI 0.66 to 1.01, *p* = 0.063). In conclusion, the administration of identical COVID-19 mRNA vaccine boosters within six months to SOT recipients may result in limited humoral immunogenicity of the last dose.

## 1. Introduction

After infection with SARS-CoV-2 solid organ transplant (SOT), recipients remain at risk of severe outcomes due to COVID-19 [1,2,3]. Although SOT recipients elicit inferior responses to COVID-19 mRNA vaccines compared to healthy populations [4,5,6,7], vaccination is an important preventive measure to lower the risk of severe disease in SOT recipients [2,8,9,10,11,12,13,14].

Although no protective antibody concentration threshold has been established, lower SARS-CoV-2 antibody concentration has previously been reported to be associated with increased risk of severe COVID-19 [15,16]. A third primary dose and additional booster doses have been shown to improve humoral immune responses and vaccine effectiveness in SOT recipients [4,8,13,14,17,18,19,20,21,22,23]. Furthermore, booster doses have been recommended as both humoral immunity and the effectiveness of COVID-19 mRNA vaccines wane over time [5,7,24]. Hence, many SOT recipients have been vaccinated with multiple COVID-19 mRNA vaccine boosters. In Denmark, most SOT recipients have received monovalent BNT162b2 vaccines as the first four vaccine doses and bivalent BNT162b2 Original/BA.1 or BNT162b2 Original/BA.4-5 as the fifth vaccine dose.

Although the WHO recommends an interval between the administration of COVID-19 booster doses of 6–12 months in immunocompromised individuals [25], Australian health authorities recommend COVID-19 vaccination of immunocompromised individuals every six months [26], and the Centers for Disease Control (CDC) recommends at least a two-month interval between COVID-19 booster doses [27]. Evidence on the impact of the interval between boosting with COVID-19 mRNA vaccines on humoral responses in SOT recipients is limited to studies indicating a beneficial effect of an extended interval between the first and second vaccine dose and second and third vaccine dose in immunocompetent populations [28,29,30,31,32]. Depending on the timing of both future SARS-CoV-2 surges and modifications of the available COVID-19 mRNA vaccines, repeated vaccination with identical epitopes might become relevant again in the future. As most SOT recipients have received a primary vaccine series of three or more vaccines, we aimed to investigate the impact of time between identical COVID-19 mRNA booster vaccine doses on humoral vaccine responses among SOT recipients previously vaccinated with three or more doses of BNT162b2. We hypothesized that a shorter interval between booster doses impacts humoral responses negatively.

## 2. Methods

### 2.1. Study Design

This study is a prospective observational cohort study of immune responses to SARS-CoV-2 infections and vaccinations in SOT recipients followed at Copenhagen University Hospital, Rigshospitalet. Vaccination against SARS-CoV-2 in Denmark was initiated on 27 December 2020. From January 2021 through April 2021, adult kidney, liver, and lung transplant recipients who had not yet received their second SARS-CoV-2 vaccine dose were invited to participate. From July 2021, adult kidney, liver, and lung transplant recipients were invited to participate regardless of vaccination status. Blood was collected at predefined times at study entry, approximately 3 weeks, 2 months, 6 months, 12 months, 18 months, and 24 months after the first SARS-CoV-2 vaccine dose, regardless of the administration of additional SARS-CoV-2 vaccine doses. The results from the first 12 months of the study have previously been described [5,6].

In this study, we used results from samples collected at 18 months and 24 months. Since we aimed to study the impact of time between identical COVID-19 mRNA vaccine booster doses on humoral vaccine responses among SOT recipients previously vaccinated with three or more doses of BNT162b2, the 18-month samples were defined as the baseline, and 24-month samples were defined as follow-up in the present study. To be included, participants had to have provided both a baseline and follow-up sample and have been vaccinated with a monovalent or bivalent BNT162b2 vaccine between baseline and follow-up (Figure 1). Participants with fewer than three vaccine doses before the baseline and participants vaccinated with any other COVID-19 vaccine than BNT162b2 at any time prior to the follow-up sample were excluded.

All participants provided written and oral informed consent. The study was conducted in accordance with the declaration of Helsinki and approved by the Regional Scientific Ethics Committee of the Capital Region of Denmark (H-20079890).

### 2.2. Clinical Information

Data on demographics, transplantation-related variables, medication, comorbidities, and acute graft rejections were collected from medical records. Vaccination information was collected from the Danish Vaccination Register (DDV) [33] (since 2015, it has been mandatory to register all vaccines administered in Denmark in DDV). Data on SARS-CoV-2 PCR tests were collected from the Danish Microbiology Database (MiBa), which contains information on all SARS-CoV-2 PCR tests conducted in the primary sector, hospitals, and SARS-CoV-2 test centers in Denmark [34].

### 2.3. Definitions

Participants were defined as infected before baseline if they had nucleocapsid (N) antibody-positive blood samples and/or positive SARS-CoV-2 PCR test at or before baseline.

Participants were defined as infected between baseline and follow-up if they had N-antibody-negative samples at baseline and N-antibody-positive samples at follow-up and/or positive SARS-CoV-2 PCR tests between baseline and follow-up. Participants without any N-antibody-positive samples and without any positive SARS-CoV-2 PCR tests were defined as infection-naïve.

The time between the last two BNT162b2 vaccine doses was defined as the number of days from the last previous BNT162b2 vaccine dose to the BNT162b2 vaccine dose administered between baseline and follow-up. If the interval was more than 183 days, participants were defined as having a more than six months interval between the last two vaccine doses. The time between the last two BNT162b2 vaccine doses was dichotomized as more or less than six months to reflect the WHO recommendation of at least a six-month interval between booster doses in immunocompromised individuals [25].

### 2.4. Antibody Quantification

In all participants, both ancestral strain anti-SARS-CoV-2 receptor binding domain (RBD) immunoglobulin G (IgG) antibodies and ancestral strain N-antibodies were measured. The concentration of anti-SARS-CoV-2 RBD IgG antibodies in venous blood was determined using an in-house ELISA-based assay, as previously described [3,5,35,36]. The threshold of a positive IgG response was set to 225 AU/mL based on a receiver operating characteristic (ROC) curve analysis to estimate the optimal cut-off between serum from naturally infected convalescent individuals and serum from pre-pandemic individuals obtained before COVID-19 emergence [35]. The presence of N-antibodies was measured using the Elecsys^®^ Anti-SARS-CoV-2 immunoassay (Roche Diagnostics GmbH, Mannheim, Germany) on a Cobas 8000 analyzer system (Roche Diagnostics), according to the manufacturer’s instructions.

### 2.5. Statistics

Continuous independent variables were reported as medians with interquartile ranges (IQR). Categorical independent variables were reported as frequency counts and percentages. The distribution of data was assessed by quantile–quantile plots. Geometric mean concentrations (GMCs) with 95% confidence intervals (CIs) were calculated to report anti-SARS-CoV-2 RBD IgG antibody concentrations at baseline and follow-up. To test for differences in GMCs of SARS-CoV-2 anti-RBD IgG between baseline and follow-up, the paired *t*-test was used. To calculate the change in GMCs of anti-SARS-CoV-2 RBD IgG per month, linear mixed models were fitted. The dependent variable for the minimally adjusted linear mixed models was the log10-transformed anti-SARS-CoV-2 RBD IgG concentration. The independent variables were the time from baseline to follow-up, the interval between the last two BNT162b2 vaccine doses (either as a continuous variable or dichotomized as more or less than six months), and an interaction term between time from baseline to follow-up and the vaccine dose interval variable, each treated as fixed effects. The interaction term allowed us to test whether the slope of the change in the dependent variable differed between participants with more or less than six months interval between the last two BNT162b2 vaccine doses. The intercept and slope were included as random effects. In the adjusted models, we further adjusted for age, sex, time of last infection, and the use of monoclonal antibody therapy within six months by adding the variables as fixed effects. Estimates from the models were back-transformed to report the fold change in GMCs per month and the rate ratio of fold change per month between participants with an interval between the last two BNT162b2 vaccine doses of more and less than six months. To assess the impact of SARS-CoV-2 infections on the results, linear mixed models were fitted after excluding participants with SARS-CoV-2 infection between baseline and follow-up. In sensitivity analyses, we investigated whether immunosuppressive maintenance therapy modified the association between the monthly rate of change in GMCs of anti-SARS-CoV-2 RBD IgG and the time between the last two BNT162b2 vaccine doses by adding an interaction term between the interval between the last two BNT162b2 vaccine doses and maintenance immunosuppressive therapy, either defined as a calcineurin inhibitor (yes/no), antimetabolite (yes/no) or corticosteroid (yes/no). All statistical analyses were performed using R statistical software V 4.3.0 [37] and the lme4 package [38].

## 3. Results

### 3.1. Cohort Characteristics

Out of the 277 SOT recipients included in the VACCIM cohort, 107 were vaccinated with a BNT162b2 vaccine between baseline and follow-up and had blood samples available at each time point, making them eligible for inclusion in this study (Figure 2). The median age at baseline was 61.0 years (IQR 54–67), and 66 (61.7%) were male. We included 50 (46.7%) kidney transplant recipients, 38 (35.5%) liver transplant recipients, and 19 (17.8%) lung transplant recipients. The median time from transplantation to baseline samples was 7 years (IQR 4–12), and 19 (17.8%) were retransplanted before inclusion. Calcineurin inhibitors were used as immunosuppressive maintenance therapy in 95 (88.8%) participants, while 92 (86.0%) received an antimetabolite, and 73 (68.2%) received corticosteroids. Evidence of a SARS-CoV-2 infection before baseline was present in 60 (56.1%) participants, while 17 (15.9%) were infected between baseline and follow-up. There were 96 (89.7%) participants who had an interval between their last two BNT162b2 vaccine doses of more than six months, and 11 (10.3%) had an interval of less than six months between their last two BNT162b2 vaccine doses. Data on COVID-19 vaccines and other clinical characteristics are shown in Table 1.

### 3.2. GMC of Anti-SARS-CoV-2 RBD IgG

In all participants, the GMC of anti-SARS-CoV-2 RBD IgG was 5299 AU/mL (95% CI 3185–8816) at baseline and increased to 18,140 AU/mL (12,434–26,464) at follow-up (*p* < 0.001). In participants with an interval of more than six months between their last two BNT162b2 vaccine doses, the GMC of anti-SARS-CoV-2 RBD IgG increased from 5375 AU/mL (95% CI 3105–9307) at baseline to 20,282 AU/mL (95% CI 13,689–30,050) at follow-up (*p* < 0.001), while we did not observe a statistically significant difference between baseline and follow-up GMCs of anti-SARS-CoV-2 RBD IgG in participants with an interval of less than six months between their last two doses of BNT162b2 vaccine (4680 AU/mL, 95% CI 1037–21,108 to 6849 AU/mL, 95% CI 1720–27267, *p* = 0.484) (Figure 3).

### 3.3. Monthly Change in the GMC of Anti-SARS-CoV-2 RBD IgG

To assess the impact of time from the last vaccine dose on the rate of change in the GMC of anti-SARS-CoV-2 RBD IgG per month from baseline to follow-up, we fitted linear mixed models.

In the adjusted analysis, we found a 1.34-fold change in GMCs per month (95% CI 1.26–1.44) in participants with more than six months interval between their last two BNT162b2 vaccine doses, while we found a 1.09-fold change in GMCs per month (95% CI 0.89–1.34) in participants with less than six months interval between their last two BNT162b2 vaccine doses (Figure 4 and Table 2). The monthly rate of change in participants with less than six months interval between the last two BNT162b2 vaccine doses was 0.82 times (95% CI 0.66–1.01, *p* = 0.063) the monthly rate of change in participants with more than six months interval between their last two BNT162b2 vaccine doses (Figure 5).

After excluding participants who were infected between baseline and follow-up, the monthly rate of change in GMCs in participants with less than six months interval between their last two BNT162b2 vaccine doses was 0.79 times (95% CI 0.65–0.95, *p* = 0.011) the monthly rate of change in participants with more than six months interval between their last two BNT162b2 vaccine doses (Table 2 and Figure 5). When fitting models with the interval between the last two BNT162b2 vaccine doses as a continuous variable, we found that each 1-month increment in the duration of time between the last two BNT162b2 vaccine doses was associated with an increase in the monthly rate of change of 4% in GMCs (rate ratio: 1.04 (95% CI 1.01–1.08)/month, *p* = 0.022). After excluding participants who were infected between baseline and follow-up, we found that each one-month increment in the duration of time between the last two BNT162b2 vaccine doses was associated with an increase in the monthly rate of change of 5% in GMCs (rate ratio: 1.05 (95% CI 1.02–1.08), *p* = 0.001, Appendix A). In sensitivity analyses, we investigated whether immunosuppressive maintenance therapy modified the association found between the monthly rate of change in the GMC of anti-SARS-CoV-2 RBD IgG and the time interval between the last two BNT162b2 vaccine doses. We found no significant interactions between immunosuppressive maintenance therapy and the time interval between the last two BNT162b2 vaccine doses (Appendix A).

## 4. Discussion

In this observational cohort study, we investigated the impact of time between BNT162b2 booster doses on humoral immune responses in SOT recipients. Overall, we observed an increase in the GMC of anti-SARS-CoV-2 RBD IgG in SOT recipients from baseline to follow-up. In SOT recipients with more than six months interval between their last two BNT162b2 vaccine doses, we observed an increase in the GMC of anti-SARS-CoV-2 RBD IgG between baseline and follow-up, while we did not observe a statistically significant change in the GMC of anti-SARS-CoV-2 RBD IgG between baseline and follow-up in SOT recipients with less than six months interval between the last two BNT162b2 vaccine doses. Although, there was no statistically significant association between having a less than six-month interval between the last two BNT162b2 vaccine doses and a lower rate of monthly change in the GMC of anti-SARS-CoV-2 RBD IgG among all SOT recipients. We found the association to be statistically significant when excluding SOT recipients with SARS-CoV-2 infections between baseline and follow-up. This suggests that the administration of identical COVID-19 mRNA vaccines within an interval of less than six months may have a limited impact on humoral immune responses.

To the best of our knowledge, no other previous studies have reported on the impact of time between the administration of identical COVID-19 booster vaccine doses in SOT recipients, but an extended interval between the first and second dose of BNT162b2 has previously been reported to enhance immunogenicity in immunocompetent populations. Thus, it was found that in healthcare workers, a 6–14-week interval between the first and second BNT162b2 dose resulted in both higher concentration of circulating antibodies and higher neutralizing antibody titers compared to a conventional 3–4-week dosing interval [29]. Furthermore, Parry et al. reported peak antibody concentrations to be higher among elderly people vaccinated with an 11–12-week interval between the first two doses of BNT162b2 compared to elderly people vaccinated with a 3-week dosing interval [30], and Hall et al. found both antibody concentrations and neutralization titers to be higher in healthcare workers who received two doses of BNT162b2 with 8–16 weeks interval compared to healthcare workers who received two doses BNT162b2 with 3–6 weeks interval [28]. Shaw et al. found a 12-week dosing interval between the primary and secondary doses of BNT162b2 to result in higher antibody concentrations, higher neutralization titers, and a slower decay of antibody concentration in the following six months than a 4-week dosing interval between the first two doses of BNT162b2 in adults older than 50 years [31]. Prusinkiewicz et al. investigated the impact of time between the first, second, and third vaccine doses in a cohort of Canadian paramedics and found that a longer interval between both the first and second and the second and third vaccine doses was independently associated with increased immunogenicity [32]. Thus, the findings in other populations than SOT recipients support our results, indicating a weaker humoral response when vaccinating with a shorter interval between doses. Due to concerns about weak humoral responses and findings showing improved humoral responses with the administration of additional COVID-19 mRNA booster vaccines, SOT recipients have received multiple booster doses. However, our results indicate that many boosters within short time intervals may not be the optimal strategy to improve humoral responses.

A mechanistic explanation for improved humoral immunogenicity after longer intervals between vaccine doses could be that a longer time interval allows for more time to generate and mature B cells. Nicolas et al. compared a 3-week dosing interval to a 16-week dosing interval in healthcare workers vaccinated with mRNA vaccines and found a longer dosing interval to positively impact B-cell responses and maturity, while T-cell responses were minimally affected [39]. Although Buckner et al. studied the impact of time from SARS-CoV-2 infection to the administration of booster doses of mRNA vaccines, rather than the time between vaccine doses, it is interesting that they found an interval between infection and vaccination of less than 180 days to mute B-cell responses [40], as this may indicate that repeated antigenic exposure beyond the primary vaccination series with an interval of less than six months is ineffective in mounting a humoral immune response. In our study, most participants were infected with SARS-CoV-2 prior to baseline. Importantly, the frequency of previously infected participants and the time from the last infection did not differ between participants vaccinated with intervals of more or less than six months between the last two BNT162b2 vaccine doses in our cohort.

When considering what the optimal interval between the administration of COVID-19 mRNA vaccine booster doses is in SOT recipients, it is a major limitation that no correlate of protection against disease has been established. Thus, although our study corroborates previous evidence that a longer time interval between vaccine doses has a positive effect on humoral immune responses, the clinical implications of our findings, including the impact on protection against infection and protection against severe COVID-19, remain to be elucidated. Scenarios may occur where viral mutations cause untimely surges of new SARS-CoV-2 variants, and in such cases, early boosting to improve immunity in risk groups may be tempting. Although our study does not provide evidence against such a strategy, it does imply that the impact of time between vaccine doses should be taken into consideration together with factors such as the waning of immunity and the effectiveness of vaccines over time. Thus, our data corroborate the present WHO recommendation of a six-month interval between booster doses in immunocompromised individuals when considering how to achieve increased immunogenicity of a booster dose. However, in a high-risk population such as SOT recipients, the potential benefits to immunogenicity of booster doses from longer intervals between doses may be outweighed by the potential risks of waning immunity due to less frequent vaccinations, especially in situations with high infection rates.

Our study had some limitations. Firstly, we only report on humoral responses to the ancestral strain RBD of SARS-CoV-2, and although all vaccines administered in the present study contained mRNA encoding both the ancestral and omicron strains’ SARS-CoV-2 Spike proteins, the current circulating SARS-CoV-2 variants and current vaccines are of the omicron lineage. Thus, although the scope of the study was to report on the impact of time intervals between booster doses when vaccinating with similar epitopes (the ancestral strain), it would have added important information to also report the results of the humoral response to the variant-updated parts of the vaccines. Secondly, we lacked data on cellular responses to vaccination. Lastly, the results should be interpreted carefully due to the relatively small number of participants (n = 11, 10.3%) with an interval between vaccine doses of less than six months. Our study had strengths as well. Due to well-maintained national Danish registries and the use of an N-antibody assay, we had accurate data on the time of both vaccinations and SARS-CoV-2 infections. Furthermore, we examined the impact of following current recommendations and provided novel evidence to guide future policies.

In conclusion, administration of identical COVID-19 mRNA vaccine boosters within six months to SOT recipients may result in limited humoral immunogenicity of the last dose.

## Figures and Tables

**Figure 1 viruses-16-00860-f001:**
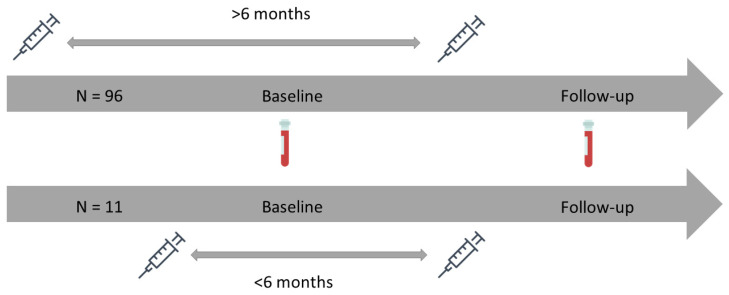
A schematic graphic showing the timeline of vaccinations and blood sampling in the study. Syringes indicate the time of vaccination and blood vials indicate the time of blood sampling.

**Figure 2 viruses-16-00860-f002:**
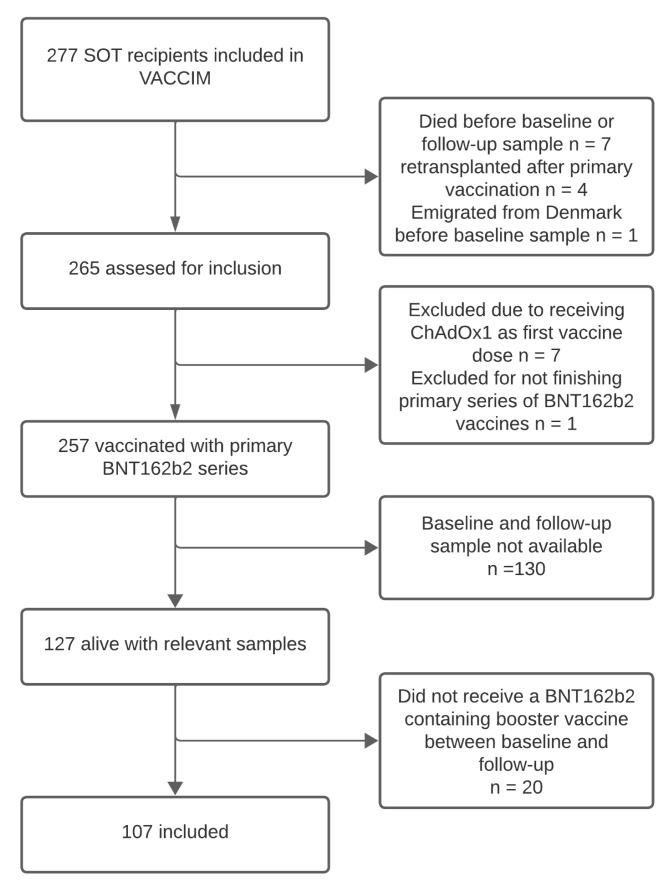
Consort diagram. A diagram showing the number of eligible SOT recipients and reasons for exclusion from the study.

**Figure 3 viruses-16-00860-f003:**
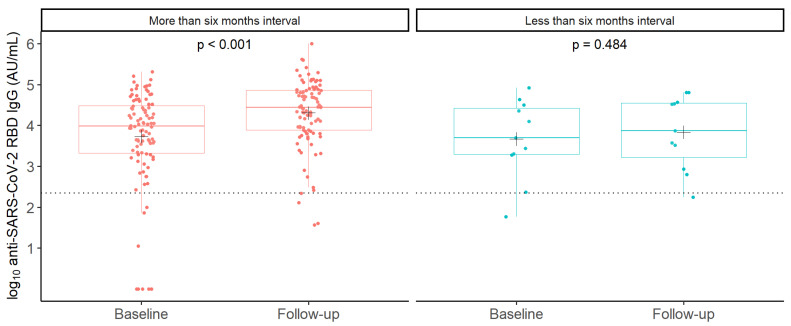
Observed anti-SARS-CoV-2 RBD IgG concentrations in SOT recipients with intervals between the last two BNT162b2 vaccine booster doses of more and less than six months. Boxplots showing median log10-transformed concentrations of anti-SARS-CoV-2 RBD IgG with interquartile ranges in SOT recipients with more (red) or less (blue) than six months interval between last two doses of BNT162b2 booster vaccine doses, before (baseline) and after (follow-up) receiving the latest booster dose. Each dot represents the observed individual concentration in each sample. Crosses (+) indicate the mean of log10-transformed anti-SARS-CoV-2 RBD IgG concentrations. A paired *t*-test was used to compare the mean log10-transformed anti-SARS-CoV-2 RBD IgG concentrations at baseline; *p*-values are printed on the figure. The dashed horizontal line indicates the assay’s minimum threshold for an IgG response.

**Figure 4 viruses-16-00860-f004:**
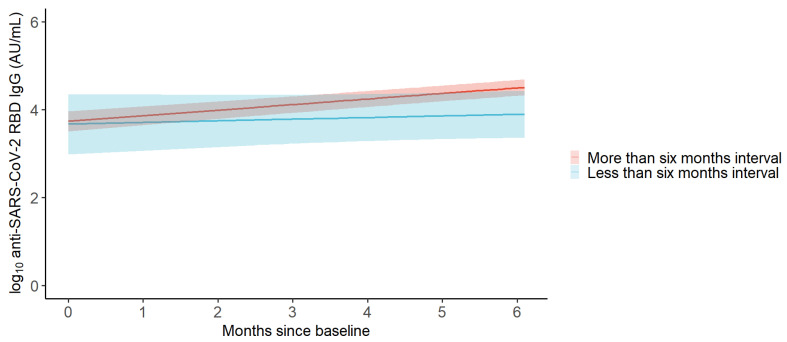
Predicted anti-SARS-CoV-2 RBD IgG concentrations in SOT recipients with intervals between their last two BNT162b2 vaccine booster doses of more and less than six months. Change in log10-transformed anti-SARS-CoV-2 RBD IgG concentration per month from baseline to follow-up in SOT recipients vaccinated with a BNT162b2 booster vaccine between baseline and follow-up as predicted by a linear mixed model. Red represents SOT recipients with an interval of more than six months between the last two BNT162b2 vaccine booster doses. Blue represents SOT recipients with less than six months interval between the last two BNT162b2 vaccine booster doses. Shaded areas represent 95% confidence intervals.

**Figure 5 viruses-16-00860-f005:**
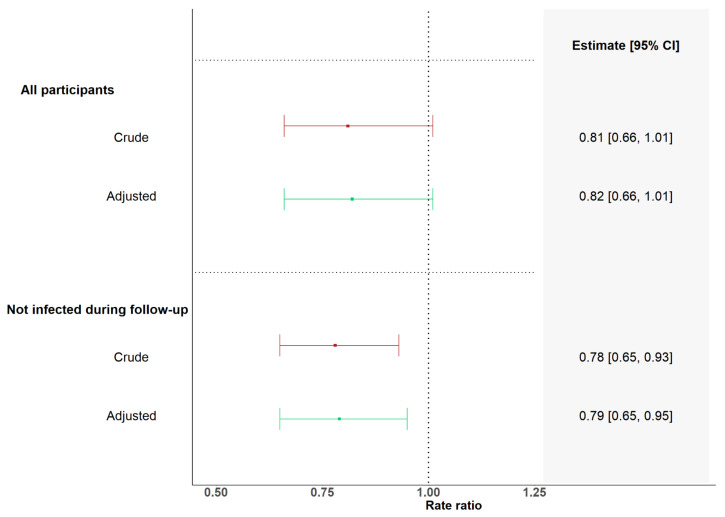
Forest plot of rate ratios between monthly fold changes in anti-SARS-CoV-2 RBD IgG concentration in SOT recipients with intervals of more or less than six months between their last two BNT162b2-containing vaccine doses. Results from linear mixed models showing rate ratios between monthly fold changes in anti-SARS-CoV-2 RBD IgG concentration in SOT recipients with intervals of more or less than six months between their last two BNT162b2 vaccine doses stratified by SARS-CoV-2 infection status. Adjustments in models containing all SOT recipients included sex, age, transplant type, infection status (infection-naïve, previous infection, and infected during follow-up), and monoclonal antibody treatment within 6 months prior to the baseline sample. Adjustments in models including SOT recipients not infected between baseline and follow-up included sex, age, transplant type, infection status (infection-naïve and previous infection), and monoclonal antibody treatment within 6 months prior to baseline sample. Adjustments in models including only infection-naïve SOT recipients included sex, age, and transplant type.

**Table 1 viruses-16-00860-t001:** Cohort characteristics.

	n = 107
Age at baseline, median [IQR] in years	61 [54, 67]
Male sex, n (%)	66 (61.7)
Time from transplantation to baseline, median [IQR] in years	7 [4, 12]
Transplant type, n (%)	
Kidney	50 (46.7)
Liver	38 (35.5)
Lung	19 (17.8)
Re-transplantation, n (%)	19 (17.8)
Comorbidities, n (%)	
Diabetes mellitus	24 (22.4)
Cardiovascular disease	81 (75.7)
Dialysis	2 (1.9)
Immunosuppressive maintenance therapy, n (%)	
Tacrolimus *	72 (67.3)
Ciclosporin *	23 (21.5)
mTOR inhibitor *	17 (15.9)
Corticosteroids	73 (68.2)
Antimetabolites	
None	15 (14.0)
Azathioprine	14 (13.1)
Mycophenolate	78 (72.9)
Time from baseline to follow-up sample, median [IQR] in months	4.5 [3.9, 4.9]
Time from baseline sample to vaccination, median [IQR] in days	35 [21, 52]
Time from vaccination to follow-up, median [IQR] in days	98 [80, 114]
Number of vaccine doses at baseline, n (%)	
3	10 (9.3)
4	92 (86.0)
5	5 (4.7)
Type of latest COVID-19 vaccine at follow-up, n (%)	
Monovalent BNT162b2	5 (4.7)
Bivalent BNT162b2/BA.1	34 (31.8)
Bivalent BNT162b2/BA.4-5	68 (63.6)
Time between last two doses of COVID-19 vaccine, median [IQR] in days	253 [241, 261]
Time between last two doses of COVID-19 vaccine, n (%)	
More than six months	96 (89.7)
Less than six months	11 (10.3)
Time of latest SARS-CoV-2 infection, n (%)	
Never infected	30 (28.0)
Before baseline	60 (56.1)
During follow-up	17 (15.9)
Monoclonal anti-SARS-CoV-2 antibody within six months before baseline, n (%)	7 (6.5)

* Six participants received both a calcineurin inhibitor (tacrolimus or ciclosporin) and an mTOR inhibitor.

**Table 2 viruses-16-00860-t002:** Monthly fold change in GMCs.

	Vaccinated with Less than Six Months Interval			Vaccinated with More than Six Months Interval			Rate Ratio of Monthly Fold Change in GMCs between SOT Recipients Vaccinated with Less than 6 Months Interval and More than 6 Months Interval		
	n	Fold Change in GMCs per Month	95% CI	n	Fold Change in GMCs per Month	95% CI	Rate Ratio	95% CI	*p*-Value
All	11			96					
Crude		1.09	0.89–1.33		1.34	1.25–1.44	0.81	0.66–1.01	0.059
Adjusted		1.09	0.89–1.34		1.34	1.25–1.44	0.82	0.66–1.01	0.063
No infection during follow-up	10			80					
Crude		0.97	0.82–1.15		1.24	1.17–1.32	0.78	0.65–0.93	0.007
Adjusted		0.99	0.83–1.17		1.25	1.18–1.33	0.79	0.65–0.95	0.011

Adjusted for age, sex, transplant type, time of last SARS-CoV-2 infection, and monoclonal SARS-CoV-2 antibody therapy within six months of baseline, as appropriate.

## Data Availability

The data are not publicly available due to privacy or ethical restrictions. The data that support the findings of this study are available upon reasonable request to the corresponding author.

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
