# Peer review of "The Impact of Time between Booster Doses on Humoral Immune Response in Solid Organ Transplant Recipients Vaccinated with BNT162b2 Vaccines"

_viruses, 2024, doi:10.3390/v16060860_

Round 1
Reviewer 1 Report
Comments and Suggestions for Authors
The authors conducted a study on the impact of time between booster doses on humoral immune response in solid organ transplant recipients vaccinated with BNT162b2 vaccines. Their findings implied that a longer interval between booster doses may lead to a more robust humoral immune response in SOT recipients, highlighting the importance of optimizing the timing of booster vaccinations in this population. Their study is helpful to improve the COVID19 immune strategy in solid organ transplant recipients.
Here are some suggestions and questions:
1. It seems that 6 months is a key time point. Immune intervals exceeding 6 months can lead to a significant increase in antibodies. What is the author's explanation for this?
2.The authors should include more discussion on the impact of immunosuppressive therapy on vaccine-induced neutralizing antibody responses.
Author Response
Please see attachtment

Reviewer 2 Report
Comments and Suggestions for Authors
The aim of this study was to investigate the effect of time between identical doses of mRNA booster vaccines against Covid-19 on the humoral response among SOT recipients previously vaccinated with three or more doses of BNT162b2. The results of the first 12 months of the study have been previously described. This study presents the results of immune status at 18 and 24 months after the first vaccination. In the study, the 18-month results were taken as baseline values and the 24-month results as control values. There is no analysis and discussion of the humoral response of patients depending on the immunosuppressant system used, as the main factor modulating the immune response in this group of patients.
Comments:
Please present in your publication a graphic - Timeline of the experiment performed, indicating the number of subjects, measurement points and vaccinations.
Line 120 – “Time between the last two BNT162b2 vaccine doses was defined as the number of days from the BNT162b2 vaccine dose administered between baseline and follow-up to last prior BNT162b2 vaccine.” Please use simple and understandable sentence syntax. It is difficult to read.
Line 139 – please add information about what constituted the independent (grouping) variable.
In the limitations of the study that may affect the conclusions, please add detailed data of a small cohort - 11 (10.3%) people with an interval of more less 6 months between their last two 167 BNT162b2 vaccine doses. There are less than 30 cases.
Limitations also include the use of a test measuring vaccine-induced antibodies, and most patients were vaccinated with Bivalent BNT162b2/BA.1 and Bivalent BNT162b2/BA.4-5 vaccines with a mutated RBD.
In "2.5. Statistics” please provide what tests were used to determine statistical differences. Please specify in this chapter what the corrections in the models were based on age, sex, time of last infection and monoclonal antibody therapy.
In point "3.1. "Cohort characteristics", please describe the immunosuppressants used.
In Table 1, it would be good to present in how many cases and what immunosuppressants were used for immunosuppression in more than one case.
It would be worth presenting the results of the humoral response of patients depending on the immunosuppressant system used.
Line 180 – please add a “statistical” difference.
Figure 2 - Please add a second X-axis at the top and appropriately separate and label the months on these axes for these 2 cohorts.
Line 193 – “[…] in each group are compared using paired t-test ns: p>0.05, **** p<0.0001.” Please correct it so it is understandable to everyone.
Line 194 – “The dashed horizontal line indicated the assays minimum threshold for an IgG response.” Please provide information on what basis this value was determined?
The introduction and discussion lack references and sources to the recommendations used in vaccination schemes, both for the basic population and for modifications used in SOT recipients.
Please rephrase the issue of the optimal recommended vaccination interval in the discussion and refer to the primary population and SOT recipients. What is more important in these populations is maintaining a high neutralization titer (more frequent interval) or higher immunogenicity of subsequent booster doses. Which option is less risky for these groups, especially SOT recipients (benefit vs. risk).
Line 293 – “Thus, although our study corroborates previous evidence that a shorter time between vaccine doses impacts humoral immune responses negatively, the clinical implications, including protection against infection and protection against severe COVID-19, of our findings remain to be elucidated.” It's probably better to point out that longer time has a positive effect - food for thought. We have no effect and a desired effect, not a negative effect.
Line 315 – “In conclusion, administration of identical COVID-19 mRNA vaccine boosters within six months to SOT recipients may have limited impact on antibody responses.” Please consider specifying instead of "antibody responses", e.g. the effectiveness of the desired effect of booster doses or the immunogenicity of booster doses, but clinical management should always refer to the specific case.
Author Response
Please see the attachtment

Round 2
Reviewer 2 Report
Comments and Suggestions for Authors
The authors took into account all comments and made appropriate corrections to the manuscript. In this form, the work meets the requirements of the Viruses magazine.